# Moral integration influences English as a Foreign Language (EFL) oral English learning: Evidence from textbook analysis and learner feedback

Xixi Yang●*, Qi Nie

Department of English, School of Foreign Languages, Jiangxi Agricultural University, Nanchang, China

* sissiyang@jxau.edu.cn

## Abstract

The research is motivated by the strategic integration of moral and value education into China's college English curriculum and the critical role of textbooks as value carriers in EFL education. Despite growing attention to value integration in EFL materials, empirical studies on moral and value representation in oral English textbooks remain sparse, and learners' perspectives on value acquisition through oral instruction are underexplored—this study aims to address these gaps. The study employed a mixed-methods approach: the researcher systematically analyzed 318 pages of *New Inside Out* Book 1 and Book 2 (first-year college EFL oral English textbooks), screening all module types including vocabulary, grammar, dialogue, listening, reading, writing, and exercise for value-related content, which was then coded and recorded in Excel. Frequency statistics show the sample textbooks include 273 value-laden verbal text samples, with a Chi-Square Test revealing statistically significant differences in dimension emphasis between the two volumes. Within Martin and White's (2005) Language Appraisal Theory, qualitative textual analysis identifies the value significance of linguistic resources as a prominent feature across module types. Additionally, to gather learner feedback on the effectiveness of value education in the oral English course, an online questionnaire survey (via Wen Juan Xing platform) was administered to learners who used the selected EFL oral English textbooks for a whole academic year. The survey showed that values-based EFL instruction positively impacted morality and personal growth.

## Introduction

In the educational landscape, the indispensable role of value education in shaping well-rounded individuals is increasingly recognized. China's Ministry of Education (MOE) (2020) emphasized, "It is an inherent requirement of talent cultivation to integrate value guidance into knowledge impartment and competence development, thereby helping students form correct worldviews, outlooks on life, and values" [1].

**Data availability statement:** All relevant data are within the paper and its Supporting Information files.

**Funding:** This paper is supported by the Educational Reform Research Project of Jiangxi Agricultural University, with the project title "Deep Reform and Practice of College English Speaking under the Background of Great Ideological and Political Education", grant number 2023B2ZZ53.

**Competing interests:** The authors have declared that no competing interests exist.

This statement precisely clarifies the core role of value education: it is not isolated from teaching, but deeply integrated with knowledge and competence development, fundamentally laying a solid ideological foundation for individual growth.

The integration of morals and values into the curriculum is deemed essential, a viewpoint that Mangubhai (2007) supports by highlighting the moral and ethical dimensions of language teaching [2]. This inclusion in a teacher's cognition, is seen as fundamental in influencing the learner behavior and objectives of language instruction within the classroom. The strategic moral and value integration into the curriculum is further underscored in the *English language education key learning area curriculum guide* (Hong Kong Curriculum Development Council, 2004) [3]. This guide not only encourages the inclusion of personal and social values such as honesty, self-esteem, and perseverance but also advocates for fostering an appreciation for equality and tolerance of differences (Feng, 2017) [4]. This approach to curriculum design reflects a deliberate effort to use education as a platform for in-depth and extensive deliberation on a wide range of values, where educational materials facilitate meaningful discussions.

Textbooks are a vehicle that provides opportunities for learners to recognize and understand cultural values, including moral values, which are obeyed and used as guidelines in social life (Pratiwi et al., 2023) [5]. Textbooks are an important learning and teaching resource, as they have the potential to play a significant role in influencing learners' worldviews as they develop their understanding of different cultures (Lu et al., 2022) [6]. They go beyond transmitting subject-specific knowledge and are instrumental in nurturing moral sensitivity and behavior by including stories of role models and their behaviors, thereby significantly influencing students' moral identities (Tse & Zhang, 2017) [7]. In China, moral education is not merely a supplementary aspect but a pivotal element intended to shape students' values, worldviews. Moral education textbooks for primary schools in China from 1999 to 2005 show an increasing emphasis on moral values such as protecting the environment, loving peace, and promoting democratic cooperation (Tse & Zhang, 2017) [7]. Sense of social responsibility, patriotism are moralized and promoted as core values in Chinese education, essential for being a good citizen (Lin & Jackson, 2023) [8].

In 2021, the National Textbook Committee issued the *Guidelines for Incorporating Content Related to 'Party Leadership' into Curricula for Primary, Secondary and Tertiary Schools* [9]*,* marking the first time that content related to party leadership has been integrated into the overall design of curriculum for all levels. It requires adhering to the correct values, strengthening overall design, and enhancing educational goals, fully leveraging the core function of textbooks in education. In line with national educational policies that have systematically integrated moral and value education across various curricula, the integration of values into the curriculum and textbooks is a strategic move to align educational goals with the moral and ethical standards set by the state. This approach is particularly evident in China. The *Guide for College English Teaching* (2020 edition) points out that college English teaching should proactively integrate socialist core values, explore rich humanistic connotations, guide students to establish correct views on life, values, and the world, and lay a solid

moral foundation for cultivating foreign language talents [10]. The national emphasis on moral education is a reflection of the government's commitment to cultivating responsible and morally conscious citizens who can contribute positively to society.

Despite the national emphasis on imparting knowledge and nurturing character in college English teaching and text-books' pivotal role as value carriers, several practical questions remain to be explored regarding the imported oral English materials commonly used in Chinese contexts. For one, what core moral values are embedded in these textbooks and how are they integrated into teaching modules, and can they truly align with China's character education objectives? For another, given that "students' development of cognition, emotions, and value systems is a phased process" (Xu & Liu, 2023) [11], do value emphases across different volumes of the same textbook series show progressive adaptations to learners? Furthermore, can learners effectively perceive and internalize the textbook-embedded values, and how does this integration impact their language learning and personal growth? As noted, studies of student-perspective textbook use are needed to verify the educational goals of foreign language courses (Xu et al., 2024) [12]. These unresolved problems highlight the necessity of the current study.

## Literature review

### Moral education

Moral education is fundamentally about nurturing character and values within individuals, extending beyond the class-room to encompass a broader spectrum of societal expectations and cultural norms (Bates, 2019; Wang, 2016) [13,14]. It has evolved to include resilience, respect for core values, and personal responsibility, now recognized as the "new three Rs" in English education, reflecting a societal push towards producing individuals capable of enhancing national competitiveness and social mobility (Bates, 2019) [13]. This aligns with the broader goal of moral education globally, which aims to educate people to become good citizens in both knowledge and morality (Phan & Phan, 2006; Bates, 2019) [13,15].

Moreover, the continuous process of moral development is emphasized, suggesting that value education is not con-fined to formal institutions but is a lifelong journey that begins at home and extends into society (Prahaladaiah, 2021) [16]. Saldivar et al. (2025) found that personal values such as achievement, self-direction, security, and hedonism significantly influence international students' university selection criteria [17]. This highlights the importance of moral education in shaping individuals' decisions, even in the context of higher education.

Value education is crucial for fostering good citizenship and social cohesion. Textbooks and educational programs, such as English for speakers of other languages (ESOL) in the United States, play a significant role in educating individu-als to become good citizens, focusing on didactic education and exposing learners to cultural values and behaviors (Feng, 2017; Herrera, 2019) [4,18]. This approach is further supported by the argument that cultural content in English language teaching materials should reflect the diversity of English as an international language, catering to learners' communica-tion needs and their own cultural contexts (Rahim & Daghigh, 2020) [19]. Additionally, value education is highlighted as a critical component in negotiating multifaceted identities in English classes, where native oriental culture and philosophical values are deeply embedded (Gao, 2020) [20]. This underscores the role of moral education in cultural identity formation and the broader implications for global citizenship.

Teachers play a pivotal role in this process, serving as moral guides or role models, and their investment in student development encompasses both the sociopolitical and the educational, shaping their behaviors in the classroom (Briggs & Kim, 2020; Kusramadhani et al., 2022) [21,23]. Coleman and Leider (2013) highlight the importance of teachers posi-tioning themselves as both educators and learners in the classroom [22]. Their self-study demonstrates that teachers can empower themselves and their students by engaging in reflective practices and curriculum design that promote per-sonal and professional growth. These concerns highlight the importance of value input in teaching practices (Mangubhai, 2007) [2].

## Other important values imparted through educational approaches

Global educational initiatives are increasingly recognizing the significance of personal development and well-being, with a focus on cultivating healthy lifestyles, positive energy, and self-confidence. Textbooks and curricula promote values such as good hygiene, regular exercise, balanced diets, and self-regulation to encourage students to take responsibility for their well-being (Feng, 2017; Pratiwi et al., 2023) [4,5]. Meanwhile, programs that emphasize autonomy, resilience, and social-emotional skills aim to enhance students' self-motivation and overall well-being by encouraging active participation in community service and personal development (Bates, 2019; Han et al., 2018; Mangubhai, 2007) [2,13,24]. Collectively, these initiatives contribute to building confident and self-aware individuals who are capable of taking initiative in their learning and social interactions.

Educational materials are playing a crucial role in fostering environmental and social responsibility among students. A core undergraduate English textbook for English majors in Chinese higher education—*Understanding Contemporary China: English Public Speaking Course* (a key textbook in the *Understanding Contemporary China textbook series*)— specifically dedicates independent chapters to themes of "Making Development People-Centered" and "Live Green, Live Better" [25]. The former chapter integrates case studies of social responsibility, such as youth participation in poverty alleviation efforts. The latter chapter guides students to reflect on environmental problems and raise protection awareness through English speaking tasks. From such involvement, the textbook not only enhances students' English public speaking competence but also deepens their understanding of environmental and social responsibility.

Educational programs worldwide are focusing on developing professional and cultural competence by emphasizing values such as responsibility, respect, law-abidingness, and cultural heritage. Curricula aim to prepare students for future careers by fostering essential social and emotional skills, promoting a sense of purpose, and encouraging personal development (Bates, 2019; Han et al., 2018; Keskin, 2013; Pratiwi et al., 2023) [5,13,24,26]. Furthermore, educational initiatives that emphasize independence and the pursuit of excellence aim to empower students to become autonomous learners and contributors to society (Liu, 2023; MOE, 2017) [27,28]. Collectively, these programs contribute to developing students' cultural awareness and preparing them to be global citizens.

Educational initiatives are fostering national and cultural pride by emphasizing values such as respect, responsibility, and national identity. These initiatives aim to enhance students' self-esteem and sense of belonging by encouraging them to take pride in their cultural heritage and contributions to society (Han et al., 2018; Liu, 2023; Pratiwi et al., 2023) [5,24,27].

## Textbook representation

China had already surpassed 390 million English learners and users by 2000 (Wei & Su, 2012) [29]. Textbooks play a central role in EFL education, serving as the primary carrier of social values and taking on multiple roles in cultural teaching, including being a teacher, map, resource, trainer, and an authority and ideology (Feng, 2017; Song, 2019) [4,30]. Their influence extends to shaping teacher beliefs and professional development within educational contexts. The transition towards global textbooks in Malaysia, which has seen the replacement of locally developed textbooks with imported ones, raises questions about their ability to convey local values and underscores challenges in promoting a national identity curriculum due to the increasing tension between local autonomy and national cohesion (Rahim & Daghigh, 2020; Ye, 2022) [19,31]. Additionally, textbooks are instrumental in implementing character education policies, as demonstrated by their use in Indonesia to inculcate values and shape student behavior (Qoyyimah, 2016) [32].

The intentional construction of cultural and moral values within English language teaching (ELT) textbooks is a complex process that requires a comprehensive approach, including various linguistic, rhetorical, and multimodal resources (Xiong, 2012; Li, 2023) [33,34]. This is underscored by initiatives like the Australian Values Education Program which highlights moral development as a core educational goal, as well as by pedagogical endeavors in Chinese colleges that interpret

Confucian ethical traditions through the lens of EFL textbook content (Lovat, 2017; Ge, 2016) [35,36]. The integration of values into the curriculum, as seen in the Indonesian EFL textbook "When English Rings a Bell," and the negotiation of teacher identity with regard to morality in non-Western contexts, such as how non-Western teachers negotiate their identity with regard to morality, complicates the representation of values in EFL materials (Kusramadhani et al., 2022; Phan & Phan, 2006) [15,23].

Research provides empirical evidence of the characteristics of moral exemplars in EFL textbooks, emphasizing the importance of diverse representations and a balanced approach to cultural teaching (Han et al., 2018; Song, 2019) [24,30]. The value dimension of English language teaching, explored through educators' reflections and the challenges faced by English language teachers in engaging with value teaching practices, highlights practical challenges and ethical considerations (Briggs & Kim, 2020; Soleimani & Lovat, 2019) [21,37]. The ideological nature of language textbooks, representing the dominant culture and values, particularly in the context of Chinese as a foreign language education, is a key aspect of this discourse (Wang, 2016) [14]. Furthermore, the importance of exploring moral values and behavior through the experiences of English learners in ESOL classrooms is emphasized, suggesting a student-centered approach to moral education (Herrera, 2019) [18].

While prior research has examined the integration of morals and values into EFL materials and the complexities of cultural representation, empirical studies specifically targeting oral English textbooks and the particular morals and values they convey remain sparse. Second, few studies have investigated how moral and value emphases vary across different volumes of the same EFL oral English textbook series—an oversight given that "contents and target acquisition levels of values should be classified taking students' development levels into consideration" (Keskin, 2013) [26]. Moreover, the learner's perspective on the moral values acquired through oral English instruction in an EFL setting has yet to be thoroughly explored. To address these identified gaps, the current study seeks to investigate the following research questions:

Research Question 1 (RQ 1): What are the primary morals and values incorporated in EFL oral English textbooks, and how are they integrated and presented?

Research Question 2 (RQ 2): How do different volumes within the same EFL oral English textbook series vary in their emphasis on morals and values, and how do these differences reflect distinct educational objectives or learner needs?

Research Question 3 (RQ 3): What morals and values do learners gain from EFL oral English classes, and how do these acquisitions impact their language learning process and personal growth?

## Appliable linguistics framework for analyzing verbal texts

The Language Appraisal Theory, as developed by Martin and White (2005), offers a comprehensive approach to understanding how attitudes are expressed through language [38]. This framework categorizes attitudes into three main domains: affect, judgement, and appreciation, each with its own subcategories that allow for a nuanced exploration of emotional responses, assessments of character and behavior, and evaluations of objects and performances. Affect captures the emotional response of the speaker or writer and is divided into un/happiness, in/security, and dis/satisfaction. These subcategories reflect a range of emotions from joy and sadness to feelings of safety, confidence, and the fulfillment of desires or needs. Judgement involves the assessment of actions, behaviors, or character traits. It is further divided into social esteem, which includes normality (how usual), capacity (how capable), and tenacity (how resolute), and social sanction, encompassing veracity (how truthful) and propriety (how ethical). These subcategories evaluate the social status, acceptability, morality, ethics, and reliability of individuals or groups. Appreciation, defined as the evaluation of objects, phenomena, or performances, encompasses three key aspects: reaction (relating to impact and quality), composition (concerning balance and complexity), and valuation (pertaining to worth). This category assesses the impact, structure, organization, and worth or value of entities within specific contexts. The framework allows for a detailed analysis of how speakers or writers convey their feelings, assessments, and evaluations in various contexts.

## Mixed methods

### Sampling

The corpus of analysis is derived from two imported textbooks, *New Inside Out* (Sue Kay & Vaughan Jones, 2012) — Book 1 (Pre-intermediate) and Book 2 (Intermediate) — published by Macmillan. Popular among learners worldwide, this textbook series was introduced to China by Shanghai Foreign Language Education Press. To adapt to the domestic teaching context, the press further made integrated adaptations to the series, which is now widely used in oral English classes at Chinese universities—aiming to improve the oral English proficiency of English majors and non-English majors. Each textbook contains 12 units, with one main theme and three related sub-themes per unit, and the teaching content of each unit covers seven modules: speaking, writing, reading, listening, grammar, vocabulary, and pronunciation. The teaching schedule is 8 weeks per semester, with 2 class hours per week.

The selection of moral elements in textbooks was strictly based on two national educational documents: *Guidelines for Moral Education Work in Primary and Secondary Schools* (MOE, 2017), which outlines five broad moral education contents: Ideal and Belief Education, Socialist Core Values Education, Excellent Traditional Chinese Culture Education, Ecological Civilization Education, and Mental Health Education [28]. *Guidelines for the Construction of Curriculum Ideology and Politics in Institutions of Higher Education* (MOE, 2020) adds Vocational Ideal and Professional Ethics Education as a key content for university students [1].

By cross-referencing the moral education objectives and content of these two documents (covering primary, secondary, and tertiary education stages), the researcher synthesized "the cross-stage moral education framework" (Table 1) to screen moral elements. A total of 318 pages of *New Inside Out* (Book 1 and Book 2) were read word-by-word to identify verbal texts conveying values from each module. Ultimately, 273 samples (see S1 Data) were screened from the texts of all modules in the two textbooks, containing 27 adaptive moral elements. The distribution of these moral elements within the overall moral education content framework is presented in the table below:

The original texts of the 273 samples were extracted and recorded one by one in an Excel spreadsheet. Each sample included the following information: textbook volume (Book 1/Book 2), page number, unit, module, text content, and contained moral education elements. Through this process, the original corpus of moral education elements was established.

### Population

The survey participants were 140 first-year non-English major students from the university, with a gender ratio of male to female being 1:0.78. They used *New Inside Out* (Book 1 and Book 2) to learn oral English in the two semesters of their first academic year.

**Table 1. The cross-stage moral education framework.**

| Educational Stage | Moral Education Content Dimension | Corresponding Moral Education Elements Presented in the Textbooks |
|---|---|---|
| Primary & Secondary (Foundation) | Ideal and Belief Education | confidence, pride |
| | Socialist Core Values Education | discipline, fairness, honesty and trust, kindness and friendliness, respect, responsibility, rule awareness, social awareness |
| | Excellent Traditional Chinese Culture Education | caring and love, cultural awareness, empathy, integrity, tolerance |
| | Ecological Civilization Education | admiration for nature, care for animals, environmental awareness, thrift |
| | Mental Health Education | independence, love for the healthy lifestyle, positive energy |
| University (Advanced) | Vocational Ideal and Professional Ethics Education | career ideal, creativity, hard work, professionalism, pursuit of excellence |

## Data collection

The research data consists of two parts. First, the primary part is the original textbook corpus used for text analysis: the 273 selected corpus samples were classified according to the moral education elements they contain (i.e., samples with the same moral education element were grouped together). Then, we calculated the frequency and percentage of each moral education element in each textbook volume, followed by their total frequency and total percentage across the two volumes. The recording, classification, and basic calculations of the original corpus were all completed in Excel.

The second part of the data is the learners' feedback, collected through a questionnaire survey (see S2 Data). The questionnaire was developed via the online platform Wen Juan Xing (WJX) to collect freshmen's feedback on the effectiveness of value education in the oral English course (with one academic year of textbook use). The questionnaire comprises three parts: basic information of participants (Items 1–3, e.g., grade, gender), Likert scale questions (Items 9–12, assessing attitudes towards value integration), and multiple-choice questions (Items 4–8, focusing on the implementation and impact of value-based instruction).

Prior to participation, electronic informed consent was obtained from all respondents, who were informed of the survey's purpose, procedures, data usage, and their right to voluntary withdrawal without negative consequences. The anonymous survey (with de-identified data to protect privacy) was conducted from April 18 to May 31, 2023, with weekly reminders to improve response rates.

Of the initially distributed 140 questionnaires, 154 responses were collected (surplus due to shared IP multiple submissions). After excluding 3 non-submissions and 16 invalid responses (completion time < 35 seconds), 135 valid questionnaires were retained, yielding an effective response rate of 96.43%.

## Data analysis technique

To address RQ 1, we coded the original textual corpus with reference to Martin & White's (2005) Language Appraisal Theory. The coding was conducted across three major domains—affect, judgement, and appreciation—as well as their respective subcategories. This constitutes a qualitative analysis, through which we identified the types of emotional responses, assessments of character and behavior, and evaluations of objects and performances conveyed in the samples.

To address RQ 2, two rounds of Chi-Square Tests were conducted to examine differences in moral value distribution between the two textbooks. First, frequency data of the 27 value elements were input into SPSS 26 for the initial test. However, the test failed due to dispersed frequencies across the 27 elements—68% of cells had low expected counts, resulting in insufficient statistical power.

Thus, the 27 value types were reorganized into 6 broader dimensions based on "the cross-stage moral education framework" (see Table 1). After aggregating frequencies within each dimension, the second Chi-Square Test was performed, yielding statistically significant results: test statistics (Pearson Chi-Square, Likelihood Ratio, Fisher's Exact Test) all confirmed significant differences in moral dimension distribution between the two textbooks ($p < .05$), indicating distinct value emphases in Book 1 and Book 2.

To answer RQ3, the 135 valid questionnaire data were analyzed via basic quantitative descriptive analysis. For Likert scale items (measuring perceived value integration effectiveness), a 4-point scale (1 = lowest, 4 = highest) was used, with predefined levels (high: 3.5–4.0; above medium: 2.5–3.4; below medium: 1.5–2.4; low: 1.0–1.4). Average scores (M) were calculated for each item, then matched to levels to summarize overall perception. For items measuring moral teaching content perception, preferred methods, and observed changes, frequency and percentage of each option were counted to identify dominant perceptions and attitudes.

In summary, this study used mixed methods for three RQs: qualitative analysis for RQ1 (text coding/classification), and quantitative methods for RQ2 (Chi-Square Test) and RQ3 (basic descriptive analysis). This "qualitative identification-quantitative verification" design ensured conclusion comprehensiveness and reliability.

## The results and analysis

### Moral contents in oral English textbook series

To objectively categorize the 27 moral value elements based on their proportional distribution, we adopted the quartile method. First, the proportions were sorted in ascending order (a statistical requirement for quartile calculation), and three key quantiles were calculated: Q1 (25th percentile) = 1.47%, Q2 (median, 50th percentile) = 2.56%, and Q3 (75th percentile) = 5.49%. Based on these quantiles, the elements were divided into three tiers from high to low frequencies (see Fig 1 for detailed proportions and frequencies).

There are 7 moral elements that fall under high-frequency values (≥5.49%): *caring and love*, *love for the healthy lifestyle*, *respect*, *environmental awareness*, *responsibility*, *rule awareness*, *kindness and friendliness*. Collectively accounting for ~56.8% of total occurrences, these values reflect the textbooks' core focus on three dimensions of the moral education framework: "Excellent Traditional Chinese Culture Education", "Socialist Core Values Education", and "Ecological Civilization Education".

The 7 high-frequency moral elements align with regional and East Asian EFL textbook moral trends, supported by existing literature. First, *environmental awareness'* high frequency reflects contemporary moral priorities, consistent with Tse & Zhang (2017) [7], who noted newer Chinese primary moral textbooks emphasize protecting the environment—validating this value's presence here and extending their findings to higher education. Next, *love for the healthy lifestyle* and *caring*

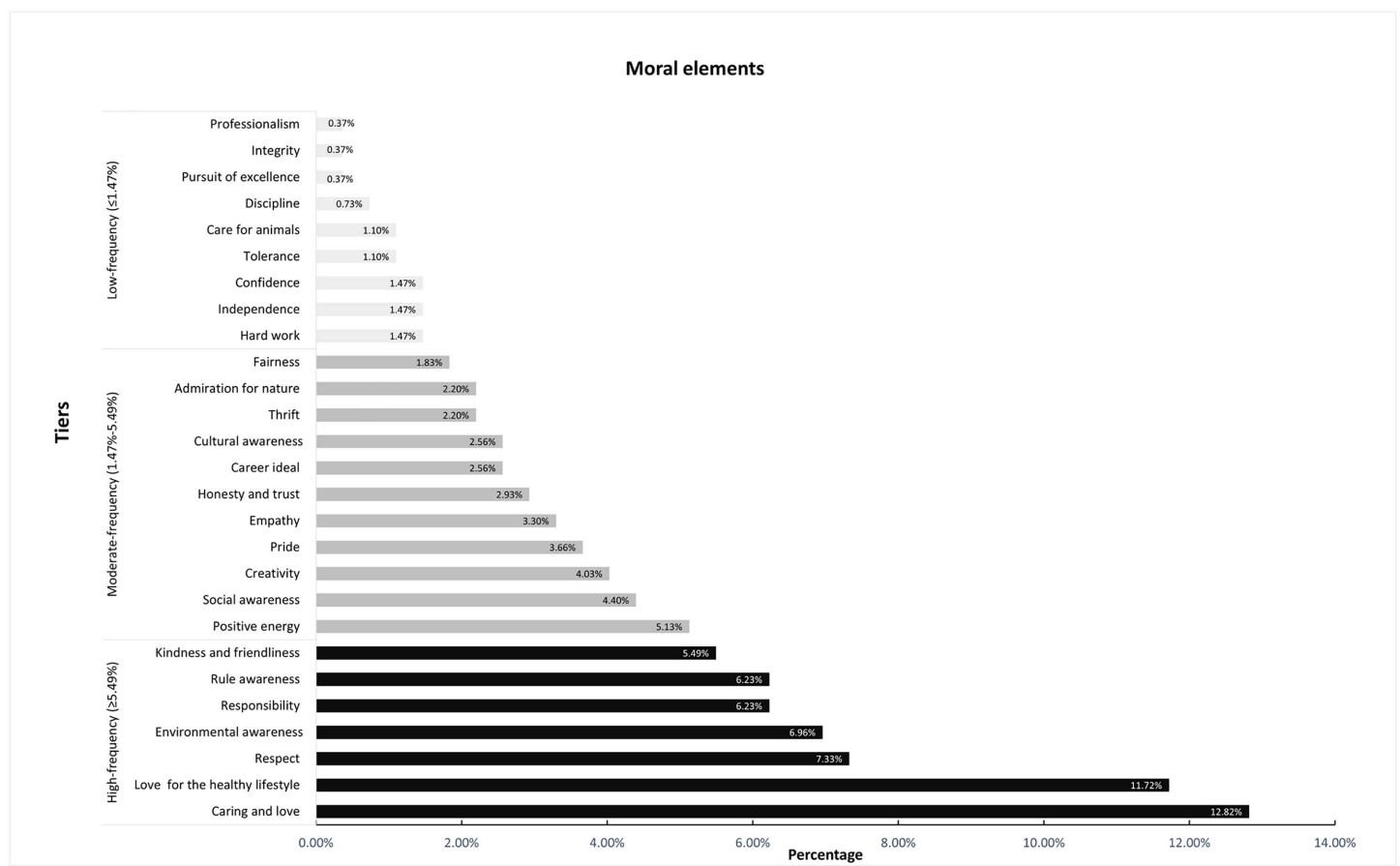

**Fig 1. Moral elements distribution in the textbook series.**

*and love* co-occur in this tier, mirroring Feng (2017) 's Hong Kong EFL textbook findings that East Asian EFL values progress from personal (e.g., healthy lifestyle) to interpersonal domains [4]. Notably, *respect* gains cross-regional support from Han et al. (2018). Their Korean-Japanese moral textbook comparison highlighted respect for human life as a shared East Asian core value [24], reinforcing that *respect's* high frequency here aligns with broader regional educational consensus.

11 moderate-frequency values (1.47% < proportion < 5.49%) include *positive energy, social awareness, creativity, pride, empathy, honesty and trust, career ideal, cultural awareness, thrift, admiration for nature, fairness*. Contributing ~35.2% to the total, this tier supplements the core values with diverse orientations and covers all moral education dimensions for primary and secondary education, with moderate representation of values emphasized at university stage (*career ideal, creativity*).

The 11 moderate-frequency elements serve as supplementary moral content, align with regional EFL textbook moral design principles, supported by the following studies. Notably, *social awareness'* moderate frequency mirrors two Indonesian findings: Kusramadhani et al. (2022) identified it as a prominent value in Indonesian junior high English textbooks (conveyed via implicit dialogues) [23], while Pratiwi et al. (2023) noted Indonesian textbooks emphasize 13 social harmony-focused themes for foreign learners [5]. This validates its presence here, reflecting cross-regional consensus on such supplementary values. For *empathy*, its inclusion echoes Bates (2019), who argued character education should prioritize recognition—empathetic connection to others [13]. Though Bates focused on English schools, the emphasis on relational skills supports why *empathy* supplements high-frequency *respect*, consistent with East Asian EFL textbooks' interpersonal focus (Feng, 2017) [4,13]. *Cultural awareness'* moderate presence gains support from two regional studies: Rahim & Daghigh (2020) found Malaysian ELT textbooks prioritize intercultural content for learners' skill needs [19], while Pratiwi et al. (2023) noted Indonesian textbooks integrate cultural values with accompanying moral dimensions into communication exercises [5]. This confirms its supplementary role, aligning with global EFL design that balances cultural literacy and language learning.

The remaining 9 moral elements belong to low-frequency values (≤1.47%): *hard work, independence, confidence, tolerance, care for animals, discipline, pursuit of excellence, integrity, professionalism*. With a total proportion of only ~8.0%, most elements belong to the framework's university-level "Vocational Ideal and Professional Ethics Education" dimension.

The low frequency of vocational values (e.g., *professionalism, pursuit of excellence*) is consistent with the textbook's freshman oral English positioning: first-year college students have minimal professional exploration needs, ensuring priority is given to language competence improvement. Next, as Tse & Zhang (2017) noted in their study, Chinese primary moral textbooks highlight hardworking attitude and loving study as core values [7], the presence of *hard work* in the current textbook further confirms that this value is an outstanding quality consistently emphasized across primary, secondary, and university education stages.

## Textbook volume disparity in value representation

Based on a total of 273 samples (Table 2), a Chi-Square Test was conducted to examine the distribution differences of moral education dimensions between Book 1 and Book 2. The results of the overall test showed that the distribution of moral education dimensions differed significantly across the two volumes (Pearson $\chi^2 = 39.593$, df = 5, $p < 0.001$; Likelihood Ratio $\chi^2 = 41.493$, df = 5, $p < 0.001$; Fisher's Exact Test = 40.439, $p < 0.001$) (Table 3).

Post hoc comparisons, derived from the Moral education dimension * volumes crosstabulation (Table 4), were based on subscript letters a and b—where different letters indicate significant differences between Book 1 and Book 2 at the 0.05 level, and the same letter indicates no significant variation. These comparisons showed that four dimensions exhibited significant between-book differences (marked with ** in Fig 2): Ecological Civilization Education (Book 1: 22[a], Book 2: 12[b]) and Mental Health Education (Book 1: 35[a], Book 2: 15[b]) were more prominent in Book 1, whereas Excellent Traditional Chinese Culture Education (Book 1: 11[a], Book 2: 44[b]) and Vocational Ideal and Professional Ethics Education (Book 1: 5[a], Book 2: 19[b]) were emphasized more in Book 2. No statistically significant differences were found in Ideal and Belief Education (Book 1: 9[a], Book 2: 5[a]) and Socialist Core Values Education (Book 1: 42[a], Book 2: 54[a]). All

**Table 2. Case Processing Summary.**

| Moral Education Dimensions * Volumes | Valid | | Missing | | Total | |
|---|---|---|---|---|---|---|
| | N | Percent | N | Percent | N | Percent |
| | 273 | 100.0% | 0 | 0.0% | 273 | 100.0% |

**Table 3. Chi-square test.**

| | Value | df | Asymptotic Significance (2-sided) | Exact Sig. (2-sided) |
|---|---|---|---|---|
| Pearson Chi-Square | 39.593[a] | 5 | .000 | .000 |
| Likelihood Ratio | 41.493 | 5 | .000 | .000 |
| Fisher's Exact Test | 40.439 | | | .000 |
| N of Valid Cases | 273 | | | |

a. 0 cells (0.0%) have expected count less than 5. The minimum expected count is 6.36.

**Table 4. Moral education dimensions * volumes crosstabulation.**

| | | | Volumes | | Total |
|---|---|---|---|---|---|
| | | | Book1 | Book2 | |
| Moral Education Dimensions | Ecological Civilization Education | Count | $22_a$ | $12_b$ | 34 |
| | | Expected Count | 15.4 | 18.6 | 34 |
| | Excellent Traditional Chinese Culture Education | Count | $11_a$ | $44_b$ | 55 |
| | | Expected Count | 25 | 30 | 55 |
| | Ideal and Belief Education | Count | $9_a$ | $5_a$ | 14 |
| | | Expected Count | 6.4 | 7.6 | 14 |
| | Mental Health Education | Count | $35_a$ | $15_b$ | 50 |
| | | Expected Count | 22.7 | 27.3 | 50 |
| | Socialist Core Values Education | Count | $42_a$ | $54_a$ | 96 |
| | | Expected Count | 43.6 | 52.4 | 96 |
| | Vocational Ideal and Professional Ethics Education | Count | $5_a$ | $19_b$ | 24 |
| | | Expected Count | 10.9 | 13.1 | 24 |
| Total | | Count | 124 | 149 | 273 |
| | | Expected Count | 124 | 149 | 273 |

Each subscript letter denotes a subset of Volumes categories whose column proportions do not differ significantly from each other at the.05 level.

expected frequencies exceeded 5 (minimum expected count = 6.36) (Table 3), meeting the prerequisite for the Chi-Square Test, and the consistency of multiple statistical indicators confirmed the reliability of the results.

## Value threads in EFL textbook narratives

In this section, the researcher looks at the narrative analysis of the representation of high-frequency values. These dominant values of the two textbook volumes are as follows: *care and love, love for the healthy lifestyle, respect, environmental awareness, responsibility, rule awareness, kindness and friendliness*. These values are prominently featured across various modules in the textbooks.

### 1) Caring and love

Text 1: Listening. I didn't want to lose him again! (Book 1 Unit 3 p20)

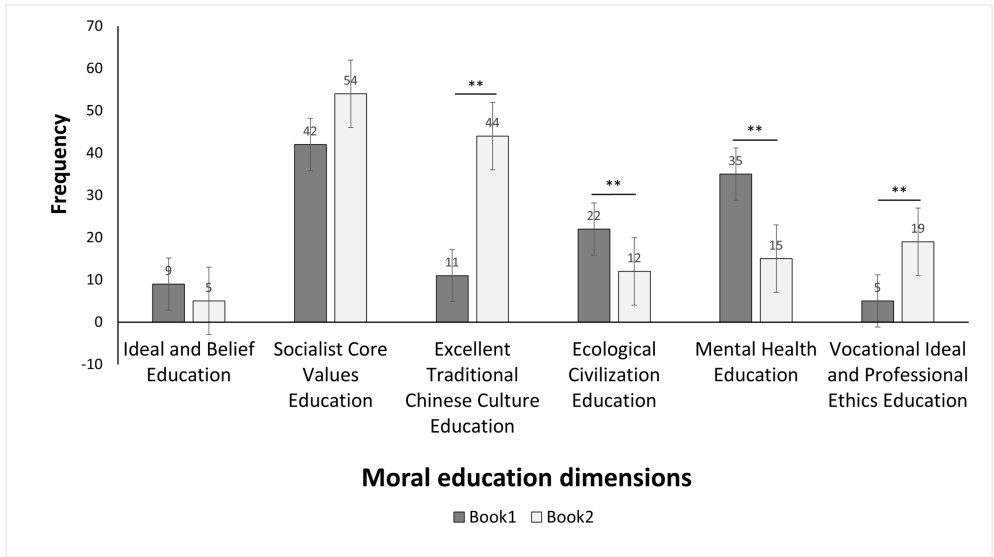

**Fig 2. Moral education dimension comparison.**

Text 2: Speaking. What's the best present you've ever given or received? (Book 1 Unit 4 p32)

Text 3: Reading. I keep this photo on my desk at work, because it makes me smile every time I look at it. It's a photo of my wife. (Book 2 Unit 3 p20)

Text 4: Grammar. You don't look very happy. Has something happened at work? (Book 2 Unit 4 p35)

Text 5: Reading & Vocabulary. A mum is a person who cares for you and tucks you in at night. (Book 2 Unit 10 p88)

The statement "I didn't want to lose him again!" in text 1(affect-dis/satisfaction) conveys a strong sense of dissatisfaction with the potential loss, reflecting an emotional attachment and the distress that comes from the possibility of losing someone cherished. This highlights the depth of the speaker's desire to maintain the relationship and the dissatisfaction that the threat of separation brings. This observes that EFL textbooks often represent social values through explicit and implicit judgments, including emotional responses like happiness, satisfaction, and distress (Feng, 2017) [4]. "Best present" in text 2 (appreciation-valuation) is a valuation of an object or gift, signifying an assessment of its worth in terms of personal experience or emotional value. Text 3 (affect-un/happiness) reflects a positive emotional response and a source of happiness, which is a subtle indication of the love and care presented in the relationship. Text 4's (judgement-social esteem-normality) inquiry about the individual's appearance and mood could be interpreted as a judgment about the normality of their behavior or emotional state. The speaker seems to recognize that the individual's current demeanor deviates from what is typically expected, prompting an inquiry that acknowledges this deviation and shows care for their well-being. Text 4 reinforces teacher's role as a moral guardian and caregiver, highlighting the importance of understanding students' emotions and supporting their psychological development (Qoyyimah et al., 2023) [39]. It also discusses the moral dimension where teachers are expected to be aware of and respond to students' emotional and psychological states (Briggs & Kim, 2020) [21]. The actions of tucking in at night and caring in text 5 (judgement-social esteem-tenacity) could be seen as indicative of a mother's tenacity and commitment to providing a sense of safety and love, which is a form of social esteem for the consistent and enduring nature of a mother's care.

### 2) Love for the healthy lifestyle

Text 6: Listening & Speaking. We all know the expression "laughter is the best medicine''. (Book 1 Unit 5 p45)

Text 7: Grammar. A mother to her teenage son: "You shouldn't watch so much television. It's bad for you.'' (Book 1 Unit 8 p73)

Text 8: Reading & Vocabulary. She gets up at 7.00 a.m. every day and goes for a brisk walk. Three times a week she plays gate-ball (a popular national sport) with her friends. There is nothing unusual about this, except that Chiako is 102 years old. (Book 1 Unit 10 p88)

Text 9: Reading. I think a happy childhood gives you balanced view of food and yourself. I eat what I fancy and feel good about it. I feel lucky that I escaped all the size zero nonsense when I was younger. (Book 2 Unit 5 p44)

The use of the word "best" in text 6 (Appreciation-valuation) indicates a positive assessment of the value of laughter in comparison to other forms of medicine or therapy. In text 7 (judgement-social sanction-propriety) the mother's advice to her son about watching less television implies a judgment on the propriety of spending excessive time on this activity, suggesting that it is not conducive to a healthy lifestyle. This advice supports the broader educational goal of promoting healthy lifestyles through self-regulation and balanced habits, as highlighted in curricula and programs aimed at personal development (Feng, 2017; Pratiwi et al., 2023) [4,5]. Text 8 (judgement-social esteem-tenacity) shows Chiako's daily routine of waking up early and engaging in physical activities, despite her age. Her commitment to these activities at the age of 102 is exceptional and shows a strong sense of determination and perseverance, which are key aspects of tenacity. In text 9 (affect-un/happiness) the speaker expresses a positive emotional state (feeling good) and a sense of luck, which are both indicators of happiness.

### 3) Respect

Text 10: Reading. What's the best way to make a really good impression at a party? By wearing the best clothes? Telling the best jokes? Dancing like a professional? No --you just need to remember people's names. (Book 1 Unit 1 p7)

Text 11: Reading & Listening. Your friend is going to meet his girlfriend's parents for the first time. Look at the advice(a-f). c)Wear the right clothes (Book 1 Unit 8 p72)

Text 12: Speaking & Reading. Don't worry about the mess! It's no fun for party guests to see the host running around holding a dustpan and brush. (Book 2 Unit 4 p36)

Text 13: Listening b) It's OK/not OK when paparazzi take unflattering photos of celebrities. (Book 2 Unit 7 p60)

Text 14: Speaking. My mother plays the piano really well, so she was very keen for me to learn the piano too. I had lessons for a while, but I was useless, and eventually my piano teacher begged my parents to stop sending me. So that was the end of my music career! But what I really wanted to do was martial arts, and my parents let me join a club when I was about nine. (Book 2 Unit 10 p93)

Text 10 (Judgement-social esteem-normality) is discussing what is considered a norm or a common practice in social interactions—that remembering people's names is a valued and effective way to make a good impression. Text 10 does resonate with the broader theme of recognizing and valuing individuals, which is demonstrated in the argument that "everyone has worth" (Mangubhai, 2007) [2]. This act of recognition can lead to positive emotions and strengthen social bonds, which is a fundamental aspect of moral education and interpersonal relationships. Wearing the right clothes in text 11 (judgement-social sanction-propriety) is advised when meeting someone's parents for the first time, which

indicates a judgment about the appropriateness of attire, reflecting social acceptability and respect for the occasion. In text 12 (judgement-social sanction-propriety) the advice not to worry about the mess and not to clean up in front of guests suggests a judgment on the propriety of hosting behavior. It implies that creating a relaxed atmosphere is more respectful to guests than obsessing over cleanliness. This emphasizes the importance of respecting others and their cultural norms, highlighting the significance of this value in fostering positive social interactions (Keskin, 2013) [26]. The statement in text 13 (judgement-social sanction-propriety) about paparazzi taking unflattering photos of celebrities touches on the propriety of such actions, questioning the ethics and respect for privacy and personal boundaries. In text 14 (affect-dis/satisfaction), to everyone's dissatisfaction, the mother's keen interest in the speaker learning piano didn't work out. The speaker uses the adjective "useless" to describe his failure. However, the speaker's real passion lies in martial arts, which the parents eventually support. This echoes the perspective that adults' values, whether parents or teachers, influence the learning climate, impacting students' personal growth and behavior (Soleimani & Lovat, 2019) [37].

## 4) Environmental awareness

Text 15: Reading. A spokesman from the Green Party said, "We are not against fashion, but cheap clothes, not designed to last, are bad for the consumer and the planet." (Book 1 Unit 4 p37)

Text 16: Listening. Debbie, 37. I'm protesting against multinational companies. They're polluting our rivers and oceans and they're causing global warming. I'm in favour of small family-run companies. I'm against food imports. I support local farmers and I buy food from farmers' markets, not supermarkets. (Book 1 Unit 7 p61)

Text 17: Reading & Vocabulary. Then imagine white sands and coral gardens never damaged by dynamite fishing or trawling nets. (Book 2 Unit 8 p69)

Text 18: Useful phrases.

Kim: Don't you think bicycles are dangerous?

Ryan: Not really. They are certainly less dangerous for the environment. (Book 2 Unit 9 p82)

The Green Party spokesman's statement in text 15 (judgement-social esteem-capacity) critiques "cheap clothes" for not being designed to last, suggesting a lack of consideration for environmental impact by the producers. This reflects a judgment on their capacity to contribute positively to sustainability. The critique of non-durable clothing is in line with the character education policy's aim to instill "environment awareness" among students, a value which is noted as part of the curriculum (Qoyyimah, 2016) [32]. This critique supports the broader educational goal of fostering environmental responsibility and sustainable practices among students, as emphasized in various educational materials and programs (Han et al., 2018; Pratiwi et al., 2023) [5,24]. Text 16 (judgement-social sanction-propriety) demonstrates her disapproval of food imports and supermarket purchases, arguing that they are ethically and socially inappropriate due to their environmental impact. Text 17 (appreciation-reaction) evokes an emotional reaction by imagining pristine white sands and coral gardens unaffected by destructive fishing practices, appreciating the untouched natural beauty and its value. This appreciation supports the broader educational goal of developing students' admiration for nature and fostering a sense of compassion and responsibility towards the environment, as highlighted in various educational initiatives (Han et al., 2018; Pratiwi et al., 2023) [5,24]. Ryan's response in text 18 (judgement-social esteem-capacity) underscores the environmental benefits of bicycles as a more sustainable and eco-friendly transportation option. This viewpoint aligns with the overarching educational objectives of cultivating environmental stewardship and promoting sustainable practices, which are integral to many contemporary educational curricula and materials (Han et al., 2018; Pratiwi et al., 2023) [5,24].

                                    

**5) Responsibility**

Text 19: Listening. I've been running a bar on the beach. It was difficult at first, because I opened my bar just two months before the tsunami in December 2004. Our bar was OK, but tourists stayed away for a long time after that, and I almost had to close the bar. But I've had a lot of support from my husband, and now the bar is going very well (Book 1 Unit 9 p80)

Text 20: Speaking. My mother did most of the cooking, but we all helped her. Well, sort of helped her. We each had a special job: I used to chop vegetables, my brother helped with the washing up, and my sister set the table. (Book 2 Unit 5 p45)

Text 21: Grammar. People have accused the fashion industry of encouraging young girls to go on starvation diets. (Book 2 Unit 7 p63)

The speaker in text 19 (judgement-social esteem-tenacity) acknowledges the great support from her husband, which was crucial during the challenging times following the tsunami. This highlights the husband's resilience and commitment to the family's well-being, demonstrating a strong sense of familial responsibility and partnership. It echoes the broader context that moral education aims to develop individuals who are responsible for their actions and can contribute positively to society (Asif et al., 2020) [40], through moral messages conveyed by the writer (Kusramadhani et al., 2022) [23]. This trait is also agreed by Text 20 (judgement-social esteem-capacity) which describes a family dynamic where each member has a "special job" in helping with cooking tasks. This allocation of roles demonstrates the family members' capacity to contribute and take responsibility for their part in the household chores. The representation of everyday roles and responsibilities within a family context is often featured by ordinary people as moral exemplars in moral education textbooks (Han et al., 2018) [24]. This approach makes moral values more relatable and attainable for students, encouraging them to emulate such behaviors in their own lives. Text 21 (judgement-social sanction-veracity) references an accusation against the fashion industry, suggesting that the speaker believes the industry is not being truthful about the potential harm to young girls' health, reflecting a lack of social responsibility.

**6) Rule awareness**

Text 22: Reading. I had to do my homework every night and I could only watch the television at the weekend. (Book 1 Unit 9 p70)

Text 23: Listening.

S: Fasten your seatbelt, grandad.

G: But I'm sitting in the back.

S: Yes, and you have to fasten your seatbelt in the back now.

G: Oh dear, all these rules and regulations. In my day, we didn't have to wear seatbelts at all. (Book 1 Unit 9 p71)

Text 24: Useful phrases. Teacher: Come along children. Stop running. OK, one by one. Ruben, stop pushing. (Book 2 Unit 6 p54)

Text 25: Reading & Listening. Schoolgirl, Pauline Gates, has not been allowed back into school after the summer holidays because she has had her nose pierced. According to headmistress, Jean Bradley, Paula knew that piercing was against the school rules. The girl will be allowed back into school when she removes the offending nosering. (Book 2 Unit 7 p64)

In text 22 (judgement-social esteem-normality) the speaker's routine of doing homework every night and watching television only on weekends reflects a normative behavior expected in a student's life. This demonstrates an acceptance of and adherence to a schedule that balances responsibilities and leisure activities, aligning with societal expectations of discipline and time management. The dialogue between the grandfather (G) and the speaker (S) in text 23 (judgement-social sanction-propriety) highlights the grandfather's resistance to the new rule of fastening seatbelts in the back seat. The speaker's insistence on following the rule, despite the grandfather's nostalgic reminiscence, underscores the societal value placed on safety and the ethical obligation to comply with current regulations. Text 24 (judgement-social sanction-propriety) shows that the teacher's instructions to the children to stop running and pushing reflect a judgment on the appropriateness of behavior, especially in a school setting. The use of commands by the teacher indicates a clear expectation for students to adhere to rules that ensure order and safety. Text 25 (judgement-social esteem-normality) describes a situation involving Pauline Gates, who has had her nose pierced against school rules, is an example of a deviation from the normative behavior expected of students. The headmistress's decision to not allow her back into school until the nose ring is removed reinforces the societal esteem for following established rules and maintaining a certain standard of appearance in educational institutions.

### 7) Kindness and friendliness

Text 26: Speaking. He's important to me because he knows me so well. When I'm feeling down or when I need to talk to somebody, I can always call him. (Book 1 Unit 1 p9)

Text 27: Reading & Speaking. There's the listener/ response smile. When two people are having a conversation, the listener smiles to encourage the speaker. (Book 1 unit 9 p76)

Text 28: Speaking & Reading. And don't forget to stock up on the chopped carrots to keep the vegetarians happy. (Book 2 Unit 4 p36)

Text 29: Reading & Speaking. Do you find it painful to share your chocolate with other people? (Book 2 Unit 5 p40)

Text 26 (judgement-social esteem-capacity) conveys that when in need, the speaker's turning to a friend hints at the underlying kindness in their relationship, suggesting a bond that provides comfort and understanding. The listener's smile in text 27 (appreciation-reaction) is a reaction to the speaker's communication, intended to have a positive impact and encourage further dialogue. Text 28 (judgement-social esteem-normality) mentions catering to vegetarians by having chopped carrots. It implies a social norm of consideration, subtly indicating an environment where kindness and respect for others' choices are valued. Text 29 reflects the perspective on intercultural teaching by demonstrating the importance of exposing students to diverse cultural practices, fostering an inclusive attitude, and promoting cultural empathy (Song, 2019) [30]. The question in text 29 (affect-un/happiness) implies a potential emotional response to the act of sharing, which could range from happiness in sharing to pain or discomfort in parting with one's chocolate.

### Learner attitudes and preferences on value integration

The questionnaire consists of three parts. Part one includes basic information about the survey participants, such as their grade and gender. Part two consists of rating scale questions, which assess students' attitudes towards the integration of values in the oral English course (Table 5). Part three includes multiple-choice questions, primarily focusing on the implementation and impact of value-based oral English instruction.

The results of the attitude part were interpreted on the basis of the four-point Likert scale (1−4): high level (3.5–4.0), above medium level (2.5–3.4), below medium level (1.5–2.4) and low level (1.0–1.4).

**Table 5. Learner perception of value integration.**

| Questions | Mean | Level of Perception |
|---|---|---|
| 9.Do you think value integration would help stimulate interest in learning? | 2.98 | Above Medium |
| 10.Do you think value integration is beneficial for improving speaking skills? | 2.96 | Above Medium |
| 11.Do you think value integration is beneficial for students' personal growth? | 3.08 | Above Medium |
| 12. Do you think teaching morals and values in oral English positively influences students' beliefs and actions? | 3.21 | Above Medium |

There is a predominantly positive perception among participants regarding the integration of value education into learning. The average scores for the questions hover at the above medium level, with the highest score of 3.21 indicating a strong positive role of moral and value education in fostering students' values and beliefs. The scores for questions related to stimulating interest in learning (2.98) and improving speaking skills (2.96) suggest a more tempered view in these areas. However, the integration is seen as beneficial for personal growth (3.08), which is above medium level. These findings underscore the value placed on moral education as a component of a well-rounded educational experience.

The survey conducted on moral issues and personal development content (Table 6) in the oral English course reveals insightful trends regarding the values perceived by participants. The top three values identified are moral qualities (84.21%), sense of right and wrong (78.20%), and life and health (73.68%). These results indicate a strong emphasis on ethical behavior and national pride among respondents, suggesting that these themes are crucial in shaping values. In the middle range, we find patriotism (72.93%), traditional culture (69.17%) and family values (67.67%). These values highlight the importance of personal well-being and familial relationships, reflecting a balanced approach to moral education that encompasses both individual and collective aspects. On the lower end of the spectrum, values such as professional conduct (57.89%), environmental protection (54.16%), and historical anecdotes (51.13%) received comparatively less emphasis.

In Item 7 about learners' preferred teaching methods for value education by oral English teachers, the most popular choices were video watching (75.97%), topic discussion (65.58%), role-playing (61.69%), and theme-based speeches (42.21%). These engaging activities not only increased learners' interest and participation in oral English classes but also encouraged critical thinking on real issues, yielding positive moral outcomes.

Regarding changes (Item 8) observed after implementing value education in oral English classes, the top four responses were: rationally viewing cultural differences between China and the West (71.43%), gaining spiritual enlightenment on truth, goodness, and beauty (65.58%), fostering love for the Party, country, and people (61.04%), and enhancing national pride (59.09%). Additionally, 43.51% noted an increased awareness of law and discipline. The survey indicates

**Table 6. Value education content perceived by learners.**

| Options | Frequency | Percentage |
|---|---|---|
| Moral qualities | 112 | 84.21% |
| Sense of right and wrong | 104 | 78.20% |
| Life and health | 98 | 73.68% |
| Patriotism | 97 | 72.93% |
| Traditional culture | 92 | 69.17% |
| Family values | 90 | 67.67% |
| Professional conduct | 77 | 57.89% |
| Environmental protection | 72 | 54.16% |
| Historical anecdotes | 68 | 51.13% |

that students are developing a respectful and inclusive attitude towards diverse cultures and values, while also cultivating a rigorous pursuit of truth and a sense of social responsibility.

## Implications

Textbook writers should optimize freshman-oriented textbooks by retaining high-frequency foundational moral elements as core moral content, properly allocating moderate-frequency supplementary elements to enrich diversity, and reducing low-frequency elements, ensuring the element distribution aligns with the " the cross-stage moral education framework " (Table 1).

Teachers should first identify frequent moral elements within the textbook modules, then play a role in shaping values by involving students in moral-embedded language knowledge building and language skill acquisition. This helps students develop sound worldviews, outlooks on life, and right values (MOE, 2020) [1] while improving language abilities. Teachers should also pay attention to the significant differences in moral education distribution between Book 1 and Book 2 of *New Inside Out*. Specifically, when teaching Book 1, focus on its strengths in Ecological Civilization Education and Mental Health Education to design relevant oral activities; when teaching Book 2, emphasize its focus on Excellent Traditional Chinese Culture Education and Vocational Ideal and Professional Ethics Education for targeted practice. As it is noted by Xu and Liu (2023) that textbook objectives should be designed based on students' cognitive levels at corresponding stages and reflect differences in hierarchy and depth across stages [11].

Based on students' preferred choices of teaching methods in value-integrated oral English classes, teachers could center around methods such as video watching, topic discussion, role-playing, thematic speeches. Allow learners to have a deeper understanding of morals through interactive and multimodal situations.

## Limitations

While the study offers a detailed analysis of the integration of value education within EFL oral English textbooks at the college level, it is not without its limitations. Firstly, the sample is confined to a specific series of EFL oral English textbooks, which may restrict the generalizability of the findings. The analysis centered on the *New Inside Out* series, used by first-year students at a university in central China. As a result, these findings may not generalize to other educational levels or contexts. Additionally, focusing on a single textbook series limits the study's ability to fully capture the diversity of EFL materials or the multiple approaches to integrating morality and values into language learning. Thirdly, only four questions are used to investigate learner perception of value integration, which is not comprehensive enough. Future research could consider a broader range of textbook samples to enhance the representativeness and universality of the findings.

## Conclusion

The integration of moral values into college-level EFL oral English textbooks is an underexplored yet critical area, as language education increasingly serves as a vehicle for individual holistic development. This study addressed this research gap by examining the representation of moral values in a widely used EFL oral English textbook series, drawing on the Language Appraisal Theory to conduct in-depth textual analysis and statistical validation.

Several key findings emerged from the study. Firstly, moral values are ingeniously embedded in thematic and activity-based language skill modules. This emphasis reflects the shift toward holistic language teaching—one that prioritizes not only linguistic proficiency but also moral and emotional development. These values focus on seven high-frequency core values: *care and love, love for the healthy lifestyle, respect, environmental awareness, responsibility, rule awareness, kindness and friendliness*.

Secondly, the distribution of moral values differs significantly between the two textbook volumes, conforming to the developmental characteristics of the academic stage. Specifically, four moral education dimensions exhibit significant differences across the two volumes: Ecological Civilization Education, Mental Health Education, Excellent Traditional

Chinese Culture Education, and Vocational Ideal and Professional Ethics Education. In contrast, Ideal and Belief Education and Socialist Core Values Education show no statistically significant differences between the two volumes, reflecting their role as foundational values consistently emphasized across both.

What's more, the value-integrated instruction proves effective—learner questionnaire survey reveals noticeable positive feedback, with improvements in learners' personal growth, value awareness, and interpersonal skills.

Based on the study we therefore suggest that textbook writers follow "the cross-stage moral education framework" to optimize freshmen-oriented textbooks by retaining high-frequency moral elements as core values, supplementing moderate-frequency elements to enrich diversity, and reducing low-frequency elements. For teachers, adopting students' preferred methods (e.g., video watching, topic discussion, role-playing, thematic speeches) in value-integrated oral English classes is advisable, as these interactive methods help nurture learners' sense of morality and translate attitudes into tangible behaviors.

## Supporting information

**S1 Data. Moral elements frequency and cases.**
(XLSX)

**S2 Data. Questionnaire of learner perspective.**
(XLSX)

## Author contributions

**Conceptualization:** Xixi Yang.

**Data curation:** Xixi Yang.

**Formal analysis:** Xixi Yang.

**Funding acquisition:** Xixi Yang.

**Investigation:** Xixi Yang.

**Methodology:** Xixi Yang.

**Project administration:** Xixi Yang.

**Resources:** Xixi Yang.

**Software:** Xixi Yang.

**Supervision:** Xixi Yang.

**Validation:** Xixi Yang.

**Visualization:** Qi Nie.

**Writing – original draft:** Xixi Yang.

**Writing – review & editing:** Xixi Yang, Qi Nie.

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
