## [Decision Letter · Decision Letter 0]

23 Oct 2025

PONE-D-25-15954How Does Value Integration Impact EFL Oral English Learning? Textbook Analysis and Learner PerspectivePLOS ONE?

Dear Dr. Yang,

Thank you for submitting your manuscript to PLOS ONE. After careful consideration, we feel that it has merit but does not fully meet PLOS ONE’s publication criteria as it currently stands. Therefore, we invite you to submit a revised version of the manuscript that addresses the points raised during the review process.

We look forward to receiving your revised manuscript.

Kind regards,

Dawit Dibekulu, PhD

Academic Editor

PLOS ONE

 [This paper is supported by the Educational Reform Research Project of Jiangxi Agricultural University, with the project title ''Deep Reform and Practice of College English Speaking under the Background of Great Ideological and Political Education'', grant number 2023B2ZZ53.].

Additional Editor Comments (if provided):

Reviewers' comments:

Reviewer's Responses to Questions

**Comments to the Author**

1. Is the manuscript technically sound, and do the data support the conclusions?

Reviewer #1: Yes

Reviewer #2: Yes

Reviewer #3: Partly

2. Has the statistical analysis been performed appropriately and rigorously?

Reviewer #1: Yes

Reviewer #2: Yes

Reviewer #3: No

3. Have the authors made all data underlying the findings in their manuscript fully available?

Reviewer #1: Yes

Reviewer #2: Yes

Reviewer #3: Yes

4. Is the manuscript presented in an intelligible fashion and written in standard English?

Reviewer #1: Yes

Reviewer #2: Yes

Reviewer #3: Yes

Reviewer #1: It is a good manuscript because it describes the stages of a correct scientific research, which is supported by empirical data from an appropriate or representative sample size as a basis for drawing conclusions. The researcher has analyzed the data quite carefully and the conclusions that has drawn are correct.

The researcher has provided all data completely and without limitations as a basis for his research findings which are explained in this manuscript.

This manuscript is presented using correct, clear and unambiguous English standards. It's just that there are a few things that need to be improved in writing.

Things that need to be improved or corrected are:

A. In the References section:

1. Abdul Rahim, H., & Jalalian Daghigh, A. (2019) is not in this manuscript excerpt. It should be removed from the references section

2. The writing of the Volume of the articles are not italicized, Check Afif et al (2020), Bates (2019), Briggs & Kim (2020), Coleman & Leider (2013), Feng (2017), Herrera (2019), Keskin (2013), Kusramadhani et al., (2022), Li (2023), and Lin & Jackson (2022), and the rest. Almost all volumes of the article in the references are written in italics. It should be changed to upright.

3. Pratiwi (2023) in the citation should be Pratiwi et al (2023) because the author is not 1 person

4. Lin, J. C., & Jackson, L. (2022) in the citation in 2023, while in the references it is written in 2022. Need to check.

5. In the references it is written Qoyyimah, U., Singh, P., Exley, B., Doherty, C., & Agustiawan, Y. (2020) in the citation written Qoyyimah et al., (2023). Need to check what year.

6. In the references it is written Pentón Herrera, L. J. (2019). It is better to start from the surname to Herrera, P. l. J. (2019).

7. The Guide for College English Teaching (2020 edition should be included in the references

8. In the citation it is written (Lu, 2022) while in the references it is written Lu, J., Liu, Y., An, L., & Zhang, Y. (2022). In the citation it needs to be changed to (Lu et al., 2022).

9. Zhang, X., & Lütge, C. (2023) is not in the citation, it should be removed from the references.

B. The Abstract section has introduced the research topic, the number of verbal text samples, and how to analyze the research data clearly, but has not stated the reasons for the importance of conducting the research, the purpose of conducting the research, and the instruments for collecting data to obtain 273 verbal text samples. Likewise, it is also necessary to explain what the survey is like.

C. In the introduction section, it is necessary to add what problems occur related to the moral content and values of the EFL English textbooks.

D. The methodology section of this manuscript needs to be organized systematically. First, explain specifically the type of mixed method research used and the reasons for applying this type. Second, explain the population and sample or subject of the research, as well as the sampling technique (Explain the process and criteria used to select samples or participants of the research). Third, explain the tools for collecting data and the method/process of collecting the data. Fourth, explain the data analysis technique. Since this research uses a Mixed Method, it is necessary to explain which data are analyzed using quantitative analysis and which data are analyzed qualitatively.

E. The results of the study are explained in this manuscript comprehensively, this is very good. It just needs to be added with the interpretation or discussion. A good manuscript is not enough to just explain the findings, it needs to be added why the findings are like that.

F. In the limitation section, it is necessary to add the limited questions in the survey (only 4 questions) to ask learner attitudes and preferences on value integration is not comprehensive.

G. At the end of this manuscript, the implications of this study need to be added.

Reviewer #2: Dear Xixi Yang,

The revision of this manuscript is mostly found in the introduction and in the body of the manuscript.

1. Introduction: as the fundamental section in which the background knowledge, significant phenomena or issues, and the synthesizing theory and related previous studies, the author has to describe and explain the hypothesis or research questions at the end of the introduction.

2. All the sub title do not follow the rules of title, author has correct all of them.

Reviewer #3: This manuscript presents a valuable mixed-methods study examining value integration in EFL oral English textbooks and its impact on learners. The research design is rigorous, combining quantitative content analysis with qualitative textual analysis using Martin and White’s (2005) Appraisal Theory framework, supplemented by learner surveys. The study makes important contributions to understanding how moral values are embedded in language teaching materials and their perceived effects on students’ development.

However, there are a few areas where the manuscript could be further strengthened to enhance the overall impact (see the attached file).

**Do you want your identity to be public for this peer review?** For information about this choice, including consent withdrawal, please see our Privacy Policy

Reviewer #1: No

Reviewer #2: **Yes:** Saifurahman Rohi

Reviewer #3: No

---

## [Author Response · Author response to Decision Letter 1]

19 Dec 2025

I have uploaded a file of complete response to editor and reviewers to answer the points one by one. For details, please see the attached file.

---

## [Editor Report · Decision Letter 1]

22 Dec 2025

Moral Integration Influences English as a Foreign Language (EFL) Oral English Learning: Evidence from Textbook Analysis and Learner Feedback

PONE-D-25-15954R1

Dear Dr. Yang,

We’re pleased to inform you that your manuscript has been judged scientifically suitable for publication and will be formally accepted for publication once it meets all outstanding technical requirements.

Kind regards,

Dawit Dibekulu, PhD

Academic Editor

PLOS One
---

## [Editor Report · Acceptance letter]

PONE-D-25-15954R1

PLOS One

Dear Dr. Yang,

I'm pleased to inform you that your manuscript has been deemed suitable for publication in PLOS One. Congratulations! Your manuscript is now being handed over to our production team.

Kind regards,

on behalf of

Dr. Dawit Dibekulu

Academic Editor

PLOS One